# Association between Serum Oxytocin, Bone Mineral Density and Body Composition in Chinese Adult Females

**DOI:** 10.3390/medicina58111625

**Published:** 2022-11-11

**Authors:** Wei-Jia Yu, Hong-Li Shi, Xiao-Qing Wu, Yan-Ping Du, Hui-Lin Li, Wen-Jing Tang, Min-Min Chen, Xue-Mei Zhang, Liu Shen, Qun Cheng

**Affiliations:** Department of Osteoporosis and Bone Disease, Research Section of Geriatric Metabolic Bone Disease, Shanghai Geriatric Institute, Huadong Hospital Affiliated to Fudan University, Shanghai 200040, China

**Keywords:** oxytocin, bone mineral density, cross-sectional study, adults, females

## Abstract

*Background and Objectives:* Oxytocin (OT) is a neuropeptide hormone which is known for its classical effects in pregnancy and lactation. Recently, growing evidence demonstrated a close relation between OT and bone. The present study aimed to explore the relationship between OT, bone and osteoporosis risk in Chinese adult females. *Materials and Methods:* in total, 149 adult females were enrolled. The serum OT levels were measured using ELISA kits. Bone mineral density (BMD) and body composition were measured by dual-energy X-ray absorptiometry (DXA). The study subjects were divided into two groups according to their menopause status and then divided into tertiles based on their serum OT level. *Results:* Serum OT, serum estradiol and BMD at three skeletal sites were significantly higher in the premenopausal group than in the postmenopausal group (*p* < 0.001, *p* = 0.008 and *p* < 0.001, respectively). In the tertile analysis, relative to tertile 1, significant associations were found for tertile 3 for OT levels and higher BMD in the femoral neck and total hip, in both pre- and postmenopausal groups. Using logistic regression analysis, tertile 3 appeared less likely to have low-BMD osteoporosis than tertile 1 (OR = 0.257, 95% CI = 0.073, 0.910). In multivariate stepwise regression analysis, OT and total lean mass were two positive determinants of BMD in the femoral neck and total hip in the premenopausal group (adjusted R^2^ for the model = 0.232 and 0.199, respectively; both *p* < 0.001). *Conclusion:* Our study demonstrated positive associations between serum OT levels and BMD in a Chinese (non-Caucasian) population. OT appeared to be more strongly associated with hip BMD in premenopausal females. These results may suggest a protective role and potential therapeutic use of OT in osteoporosis, especially for premenopausal women.

## 1. Introduction

Osteoporosis has become a growing public issue throughout the world, causing a considerable social and economic burden [1]. This disorder is characterized by impaired bone health and a subsequent high risk of fracture. Figuring out the factors affecting bone health will possibly help to solve the global health concern [2,3]. Bone health is influenced by multiple factors. The pituitary–bone axis has become increasingly recognized in bone regulation in the last decade. Recent work indicated that pituitary hormones including growth hormone, thyroid-stimulating hormone and oxytocin (OT) might be endocrine skeletal homeostasis regulators [4,5,6]. Moreover, haploinsufficient mice for pituitary hormones or their receptors exhibited severely impaired bones but unaffected primary target organs, suggesting a higher sensitivity of bone to these hormones [7,8].

The OT–bone connection has come into focus as an emerging trend [9,10]. The establishment of OT- and oxytocin receptor (OTR)-null mice provide access to the initial exploration of OT effects on bone [11,12]. Additionally, OT was discovered to be present in abundance in osteoclasts and osteoblasts, consistent with the wide distribution of OTRs [13,14]. Furthermore, preclinical studies indicated that OT stimulates the differentiation of osteoblasts while displaying dual effects on the osteoclasts by both stimulation of osteoclast formation and inhibition of osteoclast-mediated resorption, revealing OT as a novel anabolic regulator of bone mass [15]. OT was also found to promote bone formation and inhibit bone resorption in aging female rats during the alveolar healing process [16]. Clinical data demonstrated a positive relation of the serum OT level to bone mass and an opposite relation to the osteoporosis risk in postmenopausal females [17,18]. Similar results were observed in females with anorexia nervosa. Decreased nocturnal oxytocin levels in anorexia nervosa are associated with low bone mineral density and fat mass, indicating OT involvement in energy expenditure and fat metabolism [19]. In obese rodent models, peripheral or central OT administration was found to decrease the fat mass while maintaining the lean mass [20]. The effects on body composition were also confirmed in pilot studies in humans [21,22]. Together, the findings mentioned above indicate the existence of a close correlation between OT and bone and body composition.

An analysis of the relationship between OT concentration and bone mineral density (BMD)/osteoporosis and body composition has previously been reported; however, it put major emphasis on individuals with impaired bone metabolism, such as osteoporosis, hypogonadism, etc. Limited attention has been given to healthy premenopausal females, even less to Asians. Given the diversity of the ethnic genetic backgrounds and the known differences in phenotypes and determinants of osteoporosis, the exploration of the association between OT levels and bone in the Chinese population is necessary [23,24]. Furthermore, it will possibly have potential benefits for young women, helping maintain or improve their skeletal health. Thus, in the present study, we performed an association analysis to determine whether variations in serum OT levels were associated with BMD and osteoporosis risk in a Chinese community-dwelling female population, investigating the influence of OT on bone in adult females.

## 2. Materials and Methods

### 2.1. Participants

From 2017 to 2020, a total of 178 healthy female individuals were originally recruited through poster advertisements, in the Department of Osteoporosis and Bone Disease outpatient clinic, Huadong Hospital, affiliated to Fudan University, Shanghai, China. Questionnaires were used to collect demographic and clinical information, including age, sex, height, weight, medical history, family history, physical activity, alcohol use, dairy product consumption and smoking status from each study subject. To assess the correlation between OT and menopause status, we divided the study participants into two groups: premenopause and postmenopause. The inclusion criteria for the two groups were as follows: (1) premenopausal—regular menstrual cycles in the preceding 6 months, not pregnant and no oral contraceptives; (2) postmenopausal—natural menopausal status for at least 12 months. The study exclusion criteria were as follows: (1) evidence of other metabolic and inherited bone diseases such as hyper- or hypoparathyroidism, Paget’s disease, osteomalacia, osteogenesis imperfecta; (2) endocrine diseases that would affect the bone, such as hyperthyroidism, diabetes mellitus; (3) chronic renal disease, chronic liver disease or alcoholism; (4) chronic lung disease; (5) major gastrointestinal disease such as peptic ulcer, malabsorption, chronic ulcerative colitis and any significant chronic diarrhea state; (6) any current or past use of medications that would affect the bone, such as corticosteroids, anticonvulsant therapy, bisphosphonates, aromatase inhibitors and estrogen as hormone replacement therapy; (7) autoimmune disease; (8) any neurological diseases or associated cerebrovascular accident sequelae that would affect the musculoskeletal system. Participants with ovariectomy (*n* = 8), diabetes mellitus (*n* = 11), thyroid disease (*n* = 5) and urinary calculi (*n* = 5) were excluded. The remaining 149 individuals were finally included in the study.

All study subjects signed an informed consent for inclusion in the study according to the guidelines at our institution. The study was approved by the Ethics Committee of Huadong Hospital affiliated to Fudan University.

### 2.2. Assessment of Lifestyle Factors

Lifestyle factors included physical activity, alcohol consumption, dairy intake and smoking. Information on physical activity was recorded (>3 times per week and >30 min each time). Physical activities were defined as walking, running, ball games, swimming and Tai Chi. Household duties were not included. Information on alcohol consumption was collected (average alcohol consumption ≥2 units per day). Information on dairy intake was assessed (average dairy intake ≥250 mL per day). Information on smoking status was assessed (history or current tobacco use).

### 2.3. Measurement of BMD and Body Composition

BMD (g/cm^2^) measurement of the anteroposterior lumbar spine 1–4 (L1–4) and left proximal femur, including the femoral neck and total hip, and measurement of soft-tissue body compositions, including total lean mass and total fat mass, were performed by a dual-energy X-ray absorptiometry (DXA) densitometer (Hologic Delphi-A, Bedford, MA, USA). The data were analyzed using DXA analysis software. The machine was calibrated daily. The coefficients of variation (CV) of the DXA measurements were 0.86%, 1.86% and 0.95% for the lumbar spine, femoral neck and total hip, respectively [25,26]. The CVs were 0.74% and 1.5% for the total lean mass and total fat mass, respectively [26]. Total lean mass and total fat mass were expressed in terms of weight (kg) and as a percentage of body weight, i.e., the percentage of total lean mass and the percentage of total fat mass. The BMI is defined as the body mass divided by the square of the body height (kg/m^2^).

Osteoporosis was defined as a T-score of two and a half standard deviations below the mean value in healthy young adults. This is the WHO definition [27]. The osteoporosis risk was assessed separately in L1–4, femoral neck, total hip and overall. The osteoporosis thresholds were 0.730, 0.560 and 0.670 g/cm^2^ for the three sites, respectively. Osteoporosis in any of the three sites was regarded as overall osteoporosis.

### 2.4. Measurement of OT and Biochemical Markers

For each study subject, fastened morning serum samples were collected and stored at −80 °C until the assays. For the premenopausal group, the serum samples were collected on the first or the second day of the menstrual cycle to minimize the menstrual cycle-associated variability. Serum OT was measured using ELISA kits (R&D System, Minneapolis, Minnesota, USA) according to the manufacturer’s instructions. The intra-assay and inter-assay CVs were 10.2% and 5.5%, respectively. Serum estradiol (E2) was measured using a CL (Beckman Coulter, Inc., Brea, CA, USA). The intra-assay and inter-assay CVs for E2 were 4.3% and 9.1%, respectively.

### 2.5. Statistical Analysis

The basic characteristics of the study subjects are presented as mean ± SD or frequency (percentage). The Kolmogorov–Smirnov test was performed to determine whether the variables were normally distributed. Differences in baseline characteristics between the pre- and postmenopausal groups were estimated using the Student’s t-test or the unpaired non-parametric Mann–Whitney U test. Linear regression models were constructed to estimate the association between serum OT levels and BMD, with β-coefficients and corresponding 95%confidence interval (CI). Logistic regression models were constructed to estimate the association between serum OT levels and osteoporosis, with odds ratio (OR) and corresponding 95% CI. Multivariate stepwise linear regression was performed to estimate the combined effect and determinant role of the independent factors, including OT, on the dependent variable. Statistical analyses were performed using SPSS version 20.0 (SPSS, Chicago, IL, USA). Power and sample size calculations were performed using G*Power [28]. For all analyses, *p* < 0.05 was considered significant.

## 3. Results

### 3.1. Clinical Characteristics

In total, 149 study subjects comprising 74 premenopausal and 75 postmenopausal individuals were involved in the present study. We calculated that a sample size of 149 subjects would provide a power of 0.80 at an α of 0.05 (two-sided) to detect differences between the groups. The detailed basic characteristics and serum oxytocin levels by menopausal status in a community-dwelling population in Shanghai are presented in Table 1. The participants had a mean age of 51.17 years. Regarding their lifestyle, both proportions of females who smoked and who drank alcohol were very low. More than one-third and nearly one-third of the participants performed physical exercise and consumed dairy products regularly, respectively. The BMD for the three skeletal sites was significantly higher in the premenopausal individuals. As for the body composition phenotype, the total lean mass was higher while the waist/hip ratio and the percentage of total fat mass were lower in the premenopausal group in comparison with the postmenopausal group. There were no significant differences in total fat mass and in the percentage of total lean mass. Regarding the hormone levels, the serum OT and E2 levels were significantly higher in the premenopausal group.

### 3.2. Associations between Serum OT Level and BMD

The associations between BMD in lumbar spine and proximal femur and serum OT concentration are presented in Table 2. In tertile analyses, after adjustment for the menopause status, significant associations were found for tertile 3 for OT levels and higher BMD in the femoral neck (β = 0.087, 95% CI = 0.041, 0.134) and total hip (β = 0.083, 95% CI = 0.032, 0.133), relative to tertile 1. Similar positive associations were observed in both pre- and postmenopausal groups. No significant association was found between serum OT levels and lumbar spine BMD.

### 3.3. Associations between Serum OT Level and Body Composition

The association between body composition phenotypes and serum OT concentration are presented in Table 3. After adjustment for the menopause status, the results of the tertile analyses demonstrated a marginal association for tertile 3 and a higher total lean mass (β = 1.730, 95% CI = −0.035, 3.495), relative to tertile 1. The association disappeared after the individuals were divided into two groups according to the menopause status. No significant association was found between serum OT levels and other body composition phenotypes.

### 3.4. Associations between Serum OT Level and Osteoporosis

Logistic regression analyses were conducted to reveal the association between serum OT concentration and osteoporosis (Shown in Table 4). In comparison with women in tertile 1, those in tertile 3 were less likely to have a low-BMD osteoporosis (OR = 0.257, 95% CI = 0.073, 0.910). The lower risk may be mainly attributed to values recorded for the region of the femoral neck (OR = 0.155, 95% CI = 0.028, 0.845).

### 3.5. Serum OT Is a Determinant of Variation in BMD in Premenopausal Adult Females

To further determine the contribution of OT to the variations in BMD of the three examined sites, multivariate stepwise regression analysis was performed in the pre- and postmenopausal groups, using a *p*-value of 0.05 for inclusion and of 0.10 for exclusion (Shown in Table 5). Other factors known to influence the bone were entered into the multivariate analysis of OT, including age, E2, total lean mass and total fat mass. In the premenopausal group, the results demonstrated that OT and total lean mass were two positive determinants of BMD in the femoral neck and total hip (adjusted R^2^ for the model = 0.232 and 0.199, respectively; both *p* < 0.001), among other known factors influencing BMD variation (including body composition, E2 and age). Age was the only negative determinant of BMD in the femoral neck and total hip (adjusted R^2^ for the model = 0.184 and 0.189, respectively; both *p* < 0.001) in the postmenopausal group. Total lean mass was the only positive determinant of BMD in the lumbar spine in both pre- and postmenopausal groups (adjusted R^2^ for the model = 0.062 and 0.023, respectively; *p* = 0.022 and 0.004, respectively).

## 4. Discussion

In the present study, we investigated the relationship between serum OT concentration and BMD and osteoporosis in a community-dwelling population in Shanghai. As far as we know, this is the first study to reveal the significant associations between the mentioned-above phenotypes and the serum OT levels in an Asian population.

The present study demonstrated that the serum OT concentration was positively associated with BMD in the total hip and femoral neck, which is consistent with other studies [17,18,29]. No significant association was found between serum OT level and lumbar spine 1–4. Our results also suggest a lower risk of osteoporosis with a higher serum OT. Furthermore, the prevalence of osteoporosis appeared to vary by skeletal site, and the risk was lower, yet more significant, in the femoral neck than in other sites. However, though both the cortical and the trabecular bone in the femoral neck contribute to bone strength, the proportion and contribution of the cortical bone is higher; thus, the hip is considered a cortical bone site [30]. Our study might be the first to reveal that the influence of OT on bone might differ according to the skeletal site.

Recent evidence from a preclinical study also identified OT specific role in cortical bone. Fernandes et al. found that the peripheral administration of OT in female rats in periestropause contributed to the bone mass and biomechanical response of compression in the femoral neck region, especially, it influenced the cortical bone parameters (cortical bone area, cortical thickness and percentage of cortical porosity) and the cortical expression of transcription factors for osteogenesis, while it showed less influence on trabecular bone [31]. Combining these results with those of our study, we propose that OT might exert specific effects on cortical bone. During lifetime, most of the lost bone is of cortical origin, as cortical bone composes 80% of the skeleton [32]. In addition, the fact that non-vertebral fractures at predominantly cortical sites account for the majority of all fractures and most fracture-related morbidity and mortality in old age indicates the important role of cortical bone in osteoporosis and osteoporotic fractures [32]. At present, the treatment of osteoporosis primarily targets the inhibition of cancellous bone loss rather than of cortical bone loss. The significant positive correlation between OT levels and osteoporosis-related phenotypes found in our study might provide potential evidence for a targeted therapy focused on cortical bone loss. The use of peripheral quantitative computed tomography (pQCT) for further detection of the relationship between cortical bone quality and OT levels is needed in the future.

In addition, our study demonstrated that the positive association between OT level and BMD in the premenopausal group disappeared in the postmenopausal group in the multivariate regression analysis. Sun et al. also suggested that OT does not have a major role in mediating bone loss due to ovariectomy l [33]. There are several proposed explanations for this phenomenon. First, the alteration of the OT effect on bone might be attributed to variations in the estrogen levels. Estrogen is of importance in the stimulation of gene expression and protein production of OT and OTR [34]. Estrogen promotes osteogenesis in part through its effect on OT/OTR in osteoblasts [35]. As demonstrated by Maestrini et al., the evident decrease of OT concentration after menopause, also noted in the present study, indicated the existence of a relationship between OT levels and menopause status [36]. Thus, estrogen withdrawal postmenopause might affect the maintenance and further regulation of the skeletal mass by OT. Second, bone loss in postmenopausal women is probably partially due to estrogen deficiency. Another strong factor is the presence of high levels of follicle-stimulating hormone (FSH). Randolph et al. demonstrated that bone loss during the menopausal transition is more correlated with FSH rather than with estradiol [37]. Further evidence for a direct promoting effect on osteoclast synthesis and survival exerted by FSH through the FSH receptor (FSHR) emerged [38]. Blocking FSH/FSHR could prevent bone loss by inhibiting osteoclastogenesis in vitro and further stimulating bone formation in ovariectomized mice [39]. Because of the limited effect of estrogen on bone loss after menopause, the association between OT and BMD vanished after adjustment. We did not find a significant association between serum OT and E2 levels, maybe because the study sample was not large enough or the detection method of estradiol was not enough sensitive.

OT is a neuropeptide hormone released via the posterior pituitary gland. It is primarily synthesized in large (magnocellular) neurons of the paraventricular and supraoptic nuclei of the hypothalamus. OTR is widely expressed in the ovary, testis, bone, etc., suggesting OT potential profound effects [34,40,41]. OT has been known to be involved in social behavior and energy metabolism, besides delivery and milk ejection, since 1906 [12,41]. A big role of OT in the regulation of bone has been emphasized in recent years. Preclinical results demonstrated that OT might have anabolic effects on bone directly via OTR expressed in osteoblasts and osteoclasts [14,15,42]. Moreover, peripheral oxytocin could reverse ovariectomy-induced osteopenia in mice [43]. Clinical data indicate a positive relationship between OT levels and BMD in both pre-and postmenopausal women and even in hypopituitary men [17,18,29,44]. Similar results were obtained in linear regression models across the groups in the current study. A Statistically significant correlation merely remained in the premenopausal group following multivariate analysis. Thus, the correlations might be more robust in premenopausal individuals.

The present study has several limitations that should be noted. Since it was a cross-sectional study, longitudinal studies are needed to further investigate the cause-and-effect relationships. Though the intergroup differences were significant with a power of 0.80, there might have been risk of bias due to the limited sample size. In addition, the association analysis of serum oxytocin was not performed simultaneously in male individuals to determine whether a sex-specific contribution exists. Another limitation is the lack of information about the employment status and job types of study participants, which may affect their OT concentration. A more detailed study with relevant information is needed in the future [45].

## 5. Conclusions

In conclusion, we are the first who investigated the relationship between serum OT concentration and BMD at different bone sites and the related risk for osteoporosis in a community-dwelling Asian population. We observed that OT was positively associated with BMD and negatively associated with the osteoporosis risk. In addition, OT might be a determinant of BMD in premenopausal women. Our findings indicate a role for OT as a protective factor against low bone mass in women. These findings not only enhance our knowledge of the relationship between OT, BMD and osteoporosis in different ethnicities, but also might provide new insights into unsettled issues in the regulation of cortical bone in humans by revealing OT role in it.

## Figures and Tables

**Table 1 medicina-58-01625-t001:** Basic characteristics and serum oxytocin levels by menopausal status in Chinese adult women.

	Total (*n* = 149)	Menopause Status		
		Premenopausal Women (*n* = 74)	Postmenopausal Women (*n* = 75)	*p* Values
Age (years)	51.17 ± 18.39	33.55 ± 8.75	67.89 ± 8.63	*<0.001*
BMI (kg/m^2^)	22.947 ± 2.684	22.797 ± 2.490	22.978 ± 2.897	0.754
**Lifestyle (%, *n*)**				
Physical activity	34.22 (51/149)	33.78 (25/74)	34.67 (26/75)	0.923
Alcohol consumption	1.34 (2/149)	1.35 (1/74)	1.33 (1/75)	1.000
Dairy intake	27.52 (41/149)	24.32 (18/74)	30.67 (23/75)	0.897
Smoking status	0.67 (1/149)	1.35 (1/74)	0 (0/75)	0.339
**BMD (g/cm^2^)**				
L1–4	0.898 ± 0.163	1.013 ± 0.106	0.782 ± 0.124	*<0.001*
Femoral neck	0.714 ± 0.132	0.780 ± 0.114	0.649 ± 0.116	*<0.001*
Total hip	0.846 ± 0.135	0.898 ± 0.107	0.794 ± 0.139	*<0.001*
**Body composition**				
Total fat mass(kg)	20.131 ± 4.694	19.881 ± 4.979	20.287 ± 4.402	0.612
Percentage of total fat mass(%)	33.482 ± 4.498	32.407 ± 4.653	34.518 ± 4.141	*0.006*
Total lean mass(kg)	37.692 ± 4.466	38.983 ± 4.566	36.290 ± 3.897	*<0.001*
Percentage of total lean mass(%)	65.412 ± 7.548	66.635 ± 9.333	64.136 ± 4.807	0.051
Waist-hip ratio	1.013 ± 0.160	0.917 ± 0.125	1.107 ± 0.137	*<0.001*
**Hormone levels**				
E2 (pmol/L)	78.00 (37, 195)	174 (115, 325)	37 (37,41)	*<0.001* ^a^
Oxytocin (pg/mL)	563.32 (240.73, 1072.07)	777.01 (288.48, 1145.44)	364.05 (206.20, 729.08)	*0.008* ^a^

^a^ Data were log-transformed before the statistical analysis. BMI, body mass index; BMD, bone mineral density; E2, estradiol. Italic indicates significant *p*-values (*p* < 0.05).

**Table 2 medicina-58-01625-t002:** Association between serum oxytocin levels and BMD in Chinese adult women by menopausal status.

	BMD (g/cm^2^)					
Oxytocin	L1–4 β (95% CI)	*p* Values	Femoral Neck β (95% CI)	*p* Values	Total Hip β (95% CI)	*p* Values
**Total Women ^a^**						
Tertile 1	Reference		Reference		Reference	
Tertile 2	−0.002 (−0.047, 0.044)	0.948	0.025 (−0.020, 0.069)	0.274	0.027 (−0.022, 0.076)	0.277
Tertile 3	0.040 (−0.007, 0.088)	0.094	0.087 (0.041, 0.134)	*<0.001*	0.083 (0.032, 0.133)	*0.002*
**Premenopausal women**						
Tertile 1	Reference		Reference		Reference	
Tertile 2	−0.007 (−0.065, 0.051)	0.820	0.057 (−0.006, 0.119)	0.075	0.050 (−0.008, 0.108)	0.091
Tertile 3	0.059 (−0.001, 0.119)	0.054	0.087(0.023, 0.151)	*0.009*	0.098(0.038, 0.158)	*0.002*
**Postmenopausal women**						
Tertile 1	Reference		Reference		Reference	
Tertile 2	0.022 (−0.051, 0.094)	0.553	0.063 (−0.003, 0.129)	0.063	0.057 (−0.023, 0.137)	0.158
Tertile 3	0.034 (−0.039, 0.107)	0.356	0.087 (0.021, 0.154)	*0.011*	0.090 (0.009, 0.171)	*0.030*

^a^ Linear regression was adjusted for menopausal status. BMD, bone mineral density. Italic indicates significant *p*-values (*p* < 0.05).

**Table 3 medicina-58-01625-t003:** Association between serum oxytocin level and body composition in Chinese adult women by menopausal status.

	Body Composition									
Oxytocin	Total Fat Massβ (95% CI)	*p* Values	Total Lean Mass β (95% CI)	*p* Values	Percentage of Total Fat Mass β (95% CI)	*p* Values	Percentage of Total Lean Mass β (95% CI)	*p* Values	Waist-Hip Ratio β (95% CI)	*p* Values
**Total Women ^a^**										
Tertile 1	Reference		Reference		Reference		Reference		Reference	
Tertile 2	−1.137 (−3.058, 0.784)	0.244	−0.821 (−2.527, 0.884)	0.342	−0.011 (−0.029, 0.007)	0.245	−0.009 (−0.040, 0.022)	0.569	−0.043 (−0.097, 0.010)	0.112
Tertile 3	0.649 (−1.339, 2.638)	0.520	1.730 (−0.035, 3.495)	0.055	−0.004 (−0.023, 0.015)	0.658	0.009 (−0.024, 0.041)	0.598	−0.004 (−0.059, 0.52)	0.900
**Premenopausal women**										
Tertile 1	Reference		Reference		Reference		Reference		Reference	
Tertile 2	−0.490 (−3.354, 2.373)	0.734	−0.129 (−2.792, 2.533)	0.923	−0.005 (−0.032, 0.022)	0.707	−0.020 (−0.075, 0.034)	0.460	0.002 (−0.070, 0.075)	0.945
Tertile 3	1.613 (−1.310, 4.536)	0.275	0.692 (−2.025, 3.410)	0.613	0.016 (−0.011, 0.043)	0.252	−0.015 (0.071, 0.041)	0.586	0.037 (−0.037, 0.111)	0.324
**Postmenopausal women**										
Tertile 1	Reference		Reference		Reference		Reference		Reference	
Tertile 2	−0.762 (−3.413, 1.888)	0.568	−0.523 (−2.875, 1.829)	0.659	0.006 (−0.031, 0.019)	0.626	−0.012 (−0.041, 0.016)	0.395	−0.056 (−0.138, 0.026)	0.179
Tertile 3	−1.183 (−3.860, 1.495)	0.381	0.286 (−2.090, 2.662)	0.811	−0.017 (−0.042, 0.008)	0.170	0.013 (−0.016, 0.042)	0.375	−0.024 (−0.107, 0.059)	0.568

^a^ Linear regression was adjusted for menopausal status.

**Table 4 medicina-58-01625-t004:** Association between serum oxytocin level and osteoporosis in Chinese postmenopausal women.

	Osteoporosis							
Oxytocin	L1–4 OR (95% CI)	*p* Values	Femoral Neck OR (95% CI)	*p* values	Total Hip OR (95% CI)	*p* Values	Overall OR (95% CI)	*p* Values
**Postmenopausal women**								
Tertile 1	Reference		Reference		Reference		Reference	
Tertile 2	0.450 (0.136, 1.488)	0.191	0.232 (0.052, 1.037)	0.056	0.457 (0.095, 2.202)	0.329	0.338 (0.098, 1.171)	0.087
Tertile 3	0.400 (0.118, 1.352)	0.140	0.155 (0.028, 0.845)	*0.031*	0.674 (0.154, 2.940)	0.752	0.257 (0.073, 0.910)	*0.035*

OR, odds ratio; CI, confidence interval. Italic indicates significant *p*-values (*p* < 0.05).

**Table 5 medicina-58-01625-t005:** Factors determining the variation in BMD in the multivariate stepwise regression analysis.

Dependent Variables	Independent Variables	Variables in Final Model	Standardized β	*p* Values	*p* Values for Model
**Premenopausal women**					
L1–4 BMD	Age, E2, Total lean mass, Total fat mass, Oxytocin	Total lean mass	0.275	*0.022*	*0.022*
Femoral neck BMD		Total lean mass	0.388	*0.001*	*<0.001*
		Oxytocin	0.279	*0.012*	
Total hip BMD		Total lean mass	0.340	*0.003*	*<0.001*
		Oxytocin	0.288	*0.011*	
**Postmenopausal women**					
L1–4 BMD	Age, E2, Total lean mass, Total fat mass, Oxytocin	Total lean mass	0.371	*0.004*	*0.004*
Femoral neck BMD		Age	−0.445	*<0.001*	*<0.001*
Total hip BMD		Age	−0.451	*<0.001*	*<0.001*

BMD, bone mineral density; E2, estradiol. Italic indicates significant *p*-values (*p* < 0.05).

## Data Availability

The data are not publicly available because they contain information that could compromise the privacy of the research. The datasets used and analyzed during the present study are available from the corresponding author Qun Cheng on reasonable request.

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
