# Peer review of "Association between Serum Oxytocin, Bone Mineral Density and Body Composition in Chinese Adult Females"

_medicina, 2022, doi:10.3390/medicina58111625_

Round 1
Reviewer 1 Report
Interesting article.
What is the trial registration number and the ethics number?
The introduction can be improved, please cite relevant articles such as
-Dietary patterns, body composition, and bone health in New Zealand postmenopausal women BL Ilesanmi-Oyelere, J Coad, NC Roy, MC Kruger - Frontiers in nutrition, 2020 -Vilma Clemi Colli, Roberta Okamoto, Poli Mara Spritzer, Rita Cássia Menegati Dornelles, Oxytocin promotes bone formation during the alveolar healing process in old acyclic female rats, Archives of Oral Biology -Amri, Ez-Zoubir and Pisani, Didier F.. "Control of bone and fat mass by oxytocin" Hormone Molecular Biology and Clinical Investigation, vol. 28, no. 2, 2016, pp. 95-104. https://doi.org/10.1515/hmbci-2016-0045Line 120: what is FNck? Please correct or give full meaning.
Author Response
Thank you for your excellent suggestions on our manuscript and giving us a chance to revise and improve it. We have revised the manuscript according to the reviewer’s suggestions and answered all the questions of the reviewers as follows.
Reviewer 1: Interesting article. What is the trial registration number and the ethics number? The introduction can be improved, please cite relevant articles such as -Dietary patterns, body composition, and bone health in New Zealand postmenopausal women BL Ilesanmi-Oyelere, J Coad, NC Roy, MC Kruger - Frontiers in nutrition, 2020 -Vilma Clemi Colli, Roberta Okamoto, Poli Mara Spritzer, Rita Cássia Menegati Dornelles, Oxytocin promotes bone formation during the alveolar healing process in old acyclic female rats, Archives of Oral Biology -Amri, Ez-Zoubir and Pisani, Didier F.. "Control of bone and fat mass by oxytocin" Hormone Molecular Biology and Clinical Investigation, vol. 28, no. 2, 2016, pp. 95-104. https://doi.org/10.1515/hmbci-2016-0045 Line 120: what is FNck? Please correct or give full meaning.
1) What is the trial registration number and the ethics number?
We appreciate the reviewer’s constructive comment. The ethics number of the present study is 2017K053 (shown in section Institutional Review Board Statement). Since the study is a cross-sectional, observational study without intervention or follow-up, we don’t conduct the trial registration. The ICMJE also claims that “Purely observational studies (those in which the assignment of the medical intervention is not at the discretion of the investigator) will not require registration.” (https://www.icmje.org/about-icmje/faqs/clinical-trials-registration/).
2) The introduction can be improved, please cite relevant articles such as -Dietary patterns, body composition, and bone health in New Zealand postmenopausal women BL Ilesanmi-Oyelere, J Coad, NC Roy, MC Kruger - Frontiers in nutrition, 2020 -Vilma Clemi Colli, Roberta Okamoto, Poli Mara Spritzer, Rita Cássia Menegati Dornelles, Oxytocin promotes bone formation during the alveolar healing process in old acyclic female rats, Archives of Oral Biology -Amri, Ez-Zoubir and Pisani, Didier F. "Control of bone and fat mass by oxytocin" Hormone Molecular Biology and Clinical Investigation, vol. 28, no. 2, 2016, pp. 95-104. https://doi.org/10.1515/hmbci-2016-0045
We appreciate the reviewer’s constructive comment. We have added more references including mentioned-above articles into the introduction part in the revised manuscript as suggested.
Figuring out the factors affecting bone health possibly assists to solve the global health concern [2, 3]: 2. Bolaji L Ilesanmi-Oyelere, Jane Coad, Nicole C Roy, Marlena C Kruger. Front Nutr. 2020 Oct 22;7:563689. 3. Heaney RP. Bone health. Am J Clin Nutr. 2007, 85(1),300S-3S.10.1093/ajcn/85.1.300S.
OT was also found in promotion of bone formation and inhibition of bone resorption in aging female rats during the alveolar healing process [16]: 16. Colli VC, Okamoto R, Spritzer PM, Dornelles RC. Oxytocin promotes bone formation during the alveolar healing process in old acyclic female rats. Arch Oral Biol. 2012, 57(9),1290-7.10.1016/j.archoralbio.2012.03.011.
The OT-bone connection comes into focus as an emerging trend [9, 10]: 10. Amri EZ, Pisani DF. Control of bone and fat mass by oxytocin. Horm Mol Biol Clin Investig. 2016, 28(2),95-104.10.1515/hmbci-2016-0045. 3)
Line 120: what is FNck? Please correct or give full meaning.
Thank you for the suggestion. We have corrected the “FNck” into “femoral neck”.
We would like to thank the referee again for taking time to review our manuscript.

Reviewer 2 Report
Yu et al observed serum OT, E2 and BMD alterations in Chinese adult females, and found that there is a positive correlation between serum OT levels and BMD in a Chinese (non-Caucasian) population. A few concerns about the current manuscript.
(1) In table 1, the authors claimed that there is significant difference in the BMD levels of total hip between Premenopausal women and Postmenopausal women. However, based on the data, it seems there is no significance. Please check the original data. In addition, the SPSS 13.0 version is too old for statistical analysis.
(2) In the section methods, the authors are suggested to provide a principle to divide three tertiles.
(3) The occupations of the enrolled participants should be considered and analyzed in the section discussion.
(4) The section conclusion should be more simplified. The last two sentences may be merged in the revision.
(5) Please pay attention to the rules of the abbreviations.
Author Response
Thank you for your excellent suggestions on our manuscript and giving us a chance to revise and improve it. We have revised the manuscript according to the reviewer’s suggestions and answered all the questions of the reviewers as follows.
Reviewer 2: Yu et al observed serum OT, E2 and BMD alterations in Chinese adult females, and found that there is a positive correlation between serum OT levels and BMD in a Chinese (non-Caucasian) population. A few concerns about the current manuscript.
- In table 1, the authors claimed that there is significant difference in the BMD levels of total hip between Premenopausal women and Postmenopausal women. However, based on the data, it seems there is no significance. Please check the original data. In addition, the SPSS 13.0 version is too old for statistical analysis.
We appreciate the reviewer’s constructive comment. The BMD (g/cm2) of total hip presented as mean±SD, are 0.898±0.107 and 0.794±0.139 in premenopausal group and postmenopausal group, respectively. The intergroup difference is significant at 0.001 level (shown in Table 1). We use SPSS 20.0 version to conduct statistical analysis and the results remain same.
- In the section methods, the authors are suggested to provide a principle to divide three tertiles.
We appreciate the reviewer’s constructive comment. The principles that we based on to divide three tertiles are described as follows: Firstly, the reference range for serum oxytocin level is not yet established. Secondly, oxytocin is a continuous variable that is not normally distributed, thus we divide the data into increasing percentiles and use them as dummy variables for use as predictors on the outcome. Then we can make a comparison of outcomes among participants with different levels of oxytocin. Thirdly, the sample size is not that large, thus we could not split the group into more percentiles. Therefore, we use tertile categorization of oxytocin to explore the associations of different levels of oxytocin with clinical outcomes. In addition, previous studies also used tertile categorization to analyse oxytocin (W Qian et al., 2014; Y Eisenberg et al., 2018, etc.).
- The occupations of the enrolled participants should be considered and analyzed in the section discussion.
We appreciate the reviewer’s constructive comment. The comment is valuable and helpful for improving our paper. We now include this issue as one limitation of our work, and we have added the following text to the section discussion, paragraph 6.
“Another limitation is the lack of information about the employment status and job types of study participants, which may affect the OT concentration [45]. More detailed study with relevant information is needed in the future. ”
- The section conclusion should be more simplified. The last two sentences may be merged in the revision.
Thank you for the suggestion. We have merged the last two sentences as follows:
“These findings not only enhance our knowledge of the relationship between OT and BMD and osteoporosis in different ethnicities, but also might add new insight into unsettled issues in the regulation of cortical bone in humans by revealing OT’s role in it.”
- Please pay attention to the rules of the abbreviations.
Thank you for the suggestion. According to the rules of the abbreviations, we have cancelled the abbreviation for PLM, PFM, TFM and TLM for that they are not well-known terms. We also add the full term of OR and CI when we use them for the first time in the main body.
We would like to thank the referee again for taking the time to review our manuscript.

Round 2
Reviewer 2 Report
I have no further concerns.